# Genetic diversity of *Schima superba* based on physiological traits and SSR markers

**Jiazhe Liu[1], Wenhui Shen[2,3,4], Zhangqiang Tan[2,3,4]\*, Yu Zhong[2,3,4]**

**1** Sanya Institute of Nanjing Agricultural University, Nanjing Agricultural University, Sanya, Hainan, China, **2** Guangxi Forestry Research Institute, Nanning, Guangxi, China, **3** Guangxi Laboratory of Forestry, Nanning, Guangxi, China, **4** Guangxi Key Laboratory of Superior Timber Trees Resource Cultivation, Nanning, Guangxi, China

\* 315990730@qq.com

## Abstract

*Schima superba* is an ecologically and economically valuable evergreen tree that plays a key role in reforestation, firebreak establishment, and urban landscaping in subtropical China. To evaluate its adaptive diversity, this study combined physiological trait assessment with SSR-based genetic analysis across eight natural populations comprising 122 individuals. Five physiological traits, including chlorophyll, malondialdehyde, proline, soluble protein, and soluble sugar, showed significant variation among and within populations ($p < 0.01$), with HTHL and HBB populations exhibiting the greatest phenotypic variability. Using 20 polymorphic SSR loci, we detected high genetic diversity (He = 0.804, PIC = 0.786) and moderate differentiation (Fst = 0.111) with strong gene flow (Nm = 2.23). STRUCTURE and PCoA analyses revealed five genetic clusters, and the HBN and HTHL populations displayed distinct genotypes. Mixed linear model analysis identified 14 significant SSR–trait associations, with SS30 and SS32 strongly correlated with malondialdehyde and chlorophyll content. These results demonstrate a close relationship between genetic and physiological diversity in *S. superba* and provide essential molecular resources for its conservation, breeding, and adaptive improvement.

## Introduction

*Schima superba*, commonly known as the Chinese gugertree, is an evergreen tree species with strong ecological adaptability and high economic value. It is native to subtropical regions of China [1]. Its rapid growth, adaptability, and high moisture content give it excellent fire-retardant properties, making it widely used in windbreak and firebreak plantations [2]. In addition, its straight trunk and fine-textured wood are highly valued in furniture manufacturing and construction [3]. The evergreen habit of *S. superba* also gives it high ornamental value in urban landscaping [4]. With combined functions in ecological protection, timber production, and ornamental use, *S. superba* has great potential for development and research [1].

**Data availability statement:** All relevant data are within the manuscript and its Supporting Information files.

**Funding:** This research was funded by the Central Fiscal Subsidy Project for Breeding Improved Forest Tree Varieties (Guilinbanchangzi 202503 to ZT) and the Special Fund of the National Forest Tree Germplasm Resource Collection Bank for Major Valuable Tree Species at the Guangxi Forestry Research Institute (to ZT). The funders participated in the study design, data collection and analysis, decision to publish, and preparation of the manuscript. This work was mainly completed at the corresponding author's institution, and the resulting research outputs are attributed to that institution.

**Competing interests:** The authors have declared that no competing interests exist.

**Abbreviations:** STRs, short tandem repeats; SSR, simple sequence repeats; VNTRs, variable number tandem repeats; Ho, observed heterozygosity; He, expected heterozygosity; MAS, marker-assisted selection; NA, number of alleles; PIC, polymorphic information content; PCoA, principal coordinate analysis; CHL, chlorophyll; MDA, malondialdehyde; Pro, proline; SS, soluble sugar; SP, soluble protein; TBA, thiobarbituric acid; CTAB, cetyltrimethylammonium bromide; ANOVA, analysis of variance; Fst, fixation index; Fis, inbreeding coefficient of individuals relative to the subpopulation; Fit, inbreeding coefficient of individuals relative to the total population; Nm, gene flow; Na, number of alleles; Ne, effective number of alleles; I, Shannon's information index; uHe, unbiased expected heterozygosity; HWE, Hardy–Weinberg equilibrium; MLM, mixed linear model; CV,coefficients of variation; GWAS, genome-wide association studies.

Microsatellite markers, including short tandem repeats (STRs), simple sequence repeats (SSR), and other variable number tandem repeats (VNTRs), are widely used in plant genetics because of their high polymorphism, codominant inheritance, and reproducibility. They are powerful tools for assessing genetic diversity and revealing population structure [5, 6]. *S. superba* is widely distributed, and its provenances show significant geographical differentiation, which provides a strong basis for hybrid breeding [7]. In earlier studies, 36 polymorphic SSR primers were isolated and characterized from wild populations. The Na per locus ranged from 6 to 34, with observed heterozygosity (Ho) from 0.24 to 1.00 and expected heterozygosity (He) from 0.504 to 0.945, indicating abundant genetic variation within the species [8]. More recently, transcriptome sequencing under drought stress identified more than 31,500 SSR loci from over 72,000 assembled unigenes, providing a rich resource for marker development and functional studies [9,10]. In addition, a population-level analysis using 785 high-quality SNPs from 302 accessions collected in Guangdong, Yunnan, and Guangxi showed high genetic diversity within populations (over 90% of total variance), clear regional clustering, and high gene flow (Nm = 2.9), revealing population structure patterns of value for germplasm utilization [11].

Despite these advances, several key questions remain unresolved. First, studies of physiological traits and molecular markers have mostly been conducted separately, without integrating functional traits with SSR-based genetic diversity analysis [12]. Second, although transcriptome-derived SSRs are available, few studies have systematically evaluated SSR polymorphism, population genetic structure, and their relationships with physiological traits across multiple populations [13]. Third, existing molecular diversity studies have not identified functional SSR markers associated with physiological traits that could be applied to marker-assisted selection (MAS) or targeted germplasm conservation.

Physiological traits serve as critical indicators of plant adaptation and ecological fitness [14]. Traits such as chlorophyll content, soluble proteins, soluble sugars, proline accumulation, and malondialdehyde levels reflect photosynthetic capacity, osmotic adjustment, and oxidative stress tolerance, thereby providing direct insights into plant responses to environmental variation [15]. Unlike molecular markers that mainly show genetic structure and diversity, physiological traits reveal functional differences directly related to how plants adapt and respond to their environment. Integrating physiological data with SSR-based genetic analyses therefore enables a more comprehensive evaluation of population diversity, bridging the gap between genetic variation and adaptive performance [13,16]. This combined approach has been successfully applied in other woody species, where the correlation between physiological responses and genetic diversity revealed potential markers for stress resistance and local adaptation [17,18]. For *S. superba*, such integration is particularly relevant given its ecological role in firebreaks and reforestation, where both genetic diversity and stress physiology jointly determine its survival and utility [19].

To address these gaps, this study examines 8 populations of *S. superba* using developed SSR markers to comprehensively assess genetic diversity. Genetic diversity parameters, including Ho, He, Na, and polymorphic information content (PIC),

are evaluated. Population structure is analyzed using cluster analysis and principal coordinate analysis (PCoA) and compared with geographic provenance. Finally, association analysis is performed to detect significant correlations between specific SSR genotypes and functional physiological traits, aiming to identify potential molecular markers for breeding and conservation.

By combining physiological trait data with SSR-based genetic diversity analysis, this study aims to evaluate the genetic structure of *S. superba* and uncover relationships between SSR molecular markers and physiological characteristics. The integrated approach not only enhances our understanding of how genetic variation contributes to physiological adaptability but also provides valuable guidance for germplasm conservation and the development of molecular-assisted breeding strategies for *S. superba*.

## Materials and methods

### Plant material and sampling

A total of 122 seed collections of *Schima superba* were obtained in 2011 from eight populations in Guangxi Zhuang Autonomous Region, China. These populations included Bobai (HBB, 46 individuals), Bainang Forest Farm (HBN, 3 individuals), Tianhongling Forest Farm (HTHL, 11 individuals), Qinzhou (HQZ, 31 individuals), Gongcheng (HGC, 3 individuals), Guanyang (HGY, 5 individuals), Cangwu (HCW, 8 individuals), and Rongshui (HRS, 15 individuals). The seeds were germinated and raised as seedlings in Nanning, Guangxi, and subsequently transplanted in 2013 to the experimental forest of the Guangxi Forestry Research Institute. Geographic information for the eight populations location is provided in supplementary S1 Table.

In July 2024, leaf sampling was conducted from these experimental stands. For each population, three to five healthy individuals were randomly selected, and young leaves were collected, immediately frozen in liquid nitrogen, and stored at −80 °C for genomic DNA extraction and SSR marker analysis. In addition, three to five fully expanded mature leaves were sampled from each tree for physiological measurements, with each sample set used as a biological replicate.

### Physiological characterization

Five physiological parameters were measured in all sampled individuals of *S. superba*, including chlorophyll (CHL) content, malondialdehyde (MDA) content, proline concentration (Pro), soluble sugar (SS) content, and soluble protein (SP) concentration. For each tree, six mature leaves were collected and used as biological replicates for all biochemical assays.

Chlorophyll content was determined using the acetone extraction method [20]. Malondialdehyde content was quantified using the thiobarbituric acid (TBA) method [21]. Proline concentration was assessed following the acid ninhydrin colorimetric method [22]. Soluble sugars were measured via the anthrone-sulfuric acid method [23], and soluble protein concentration was determined using the Coomassie Brilliant Blue G-250 dye-binding assay [24]. All absorbance measurements were conducted using a full-wavelength multifunctional microplate reader (Thermo Scientific, USA).

Each assay was performed using standard protocols with appropriate controls and calibration curves. Data from the six replicates per tree were averaged to obtain the final value per individual, and all individuals from each population were included in the analysis to calculate population-level means and variation.

### SSR markers and genotyping

Genomic DNA was extracted from the terminal buds of all 122 sampled individuals using the cetyltrimethylammonium bromide (CTAB) method [25]. For each sample, approximately 0.5 g of fresh leaf tissue was ground in liquid nitrogen prior to extraction. DNA quality and concentration were assessed by 1% agarose gel electrophoresis and adjusted to working concentrations of 50–200 ng/μL for downstream analyses [25].

A total of 36 SSR primer pairs previously developed for *S. superba* by Niu [14] were initially screened across eight representative populations. Based on the intensity, polymorphism, and reproducibility of the amplification products, 20 SSR markers were selected for genotyping and used for PCR amplification across all 122 individuals. Detailed information on the selected SSR primers is provided in supplementary S2 Table.

PCR amplification was performed in a 15 µL reaction volume containing 7.5 µL of 2×Taq PCR Master Mix (Tiangen Biotech, China), 1.0 µL each of forward and reverse primers, 1.0 µL of genomic DNA template, and 4.5 µL of ddH$_2$O. The PCR thermal cycling program consisted of an initial denaturation at 95 °C for 15 min, followed by 35 cycles of denaturation at 95 °C for 30 s, annealing at 56 °C for 30 s, and extension at 72 °C for 30 s, with a final extension at 72 °C for 3 min. The amplified products were separated and analyzed using capillary electrophoresis on an ABI 3130xl Genetic Analyzer (Applied Biosystems, USA), and allele sizes were determined using GeneMarker software [26].

## Statistical analysis

Nested analysis of variance (ANOVA) was performed using R to assess inter- and intra-population variation in physiological traits [27]. F-values and fixation index (Fst) were calculated to evaluate population differentiation [28]. Genetic diversity parameters including the Na, effective number of alleles (Ne), Shannon's information index (I), Ho, He, unbiased expected heterozygosity (uHe), and polymorphic information content (PIC) were calculated using GenAlEx 6.5 [29] and POPGENE 1.32 [30]. Inbreeding coefficient of individuals relative to the subpopulation (Fis), inbreeding coefficient of individuals relative to the total population (Fit) and gene flow (Nm) were also estimated. The percentage of loci deviating from Hardy–Weinberg equilibrium (HWE) was calculated per population [29]. Nei's genetic distance and genetic identity between populations were computed using POPGENE 1.32 [30]. Based on the Nei's genetic distance matrix, an UPGMA dendrogram was generated to illustrate population clustering patterns. PCoA was conducted in GenAlEx 6.5 to visualize genetic relationships among individuals [29]. Population structure was inferred using STRUCTURE 2.3.4 with the admixture model, and the optimal number of clusters (K) was determined using the ΔK method implemented in STRUCTURE HARVESTER [31]. Marker–trait associations were analyzed using a mixed linear model (MLM) in TASSEL 5.0, incorporating the Q matrix (K = 5) as a covariate to control for population structure. Associations with $p < 0.05$ were considered significant, and $R^2$ values were used to quantify the proportion of phenotypic variance explained by each locus [32].

## Results

### Variation in physiological traits among populations

To assess the physiological diversity of *S. superba*, we measured five physiological traits: CHL, MDA, Pro, SP, and SS, across eight natural populations, and summarized their mean values and coefficients of variation (CV) (Table 1). Notably, population-level differences in CV were evident. HTHL exhibited the highest overall CV (0.344), whereas HGC showed the lowest levels of intra-population variation (0.113). MDA showed particularly high variability in HTHL (CV = 0.889) and HBB (CV = 0.818).

To further analyze differences at inter- and intra-population levels, ANOVA was performed for each physiological trait (Table 2). All five traits showed statistically significant variation ($p < 0.01$) at both inter- and intra-population levels. Among them, CHL had the highest inter-population F-value (367.947), followed by SS (41.933) and Pro (13.925). SP showed relatively similar F-values at both levels, with 13.539 (inter) and 13.348 (intra), while MDA had a higher F-value at the intra-population level (12.563) than at the inter-population level (6.150). In addition, Fst values were calculated to assess the degree of population differentiation for each trait. CHL showed the highest Fst (0.766), followed by Pro (0.754) and SS (0.737), indicating relatively strong among-population divergence in these physiological parameters. MDA had the lowest Fst value (0.275), suggesting greater variation within populations than among them.

**Table 1. The mean of physiological traits and Coefficients of variation (CV) of 8 *S. superba* populations.**

| Population | CHL (mg·g$^{-1}$ FW)/CV | MDA (nmol·g$^{-1}$ FW)/CV | Pro (µg·g$^{-1}$ FW)/CV | SP (mg·g$^{-1}$ FW)/CV | SS (mg·g$^{-1}$ FW)/CV | Population CV mean |
|---|---|---|---|---|---|---|
| HQZ | 1.968/0.200 | 6.358/0.189 | 946.154/0.252 | 26.058/0.077 | 0.078/0.167 | 0.177 |
| HBB | 1.712/0.223 | 13.140/0.818 | 759.929/0.200 | 28.641/0.102 | 0.085/0.297 | 0.328 |
| HTHL | 1.762/0.270 | 11.292/0.889 | 865.372/0.156 | 26.552/0.092 | 0.085/0.315 | 0.344 |
| HRS | 2.054/0.224 | 8.552/0.313 | 730.054/0.162 | 26.837/0.184 | 0.091/0.218 | 0.220 |
| HCW | 1.778/0.256 | 7.801/0.249 | 759.925/0.178 | 28.077/0.097 | 0.094/0.194 | 0.195 |
| HGY | 1.305/0.246 | 7.002/0.146 | 591.042/0.129 | 28.231/0.109 | 0.076/0.148 | 0.155 |
| HGC | 1.277/0.074 | 6.692/0.128 | 704.128/0.198 | 26.099/0.058 | 0.064/0.110 | 0.113 |
| HBN | 2.431/0.147 | 8.482/0.207 | 594.676/0.104 | 23.444/0.095 | 0.075/0.119 | 0.134 |
| Mean traits | 1.786/0.205 | 8.665/0.367 | 743.910/0.172 | 26.742/0.102 | 0.081/0.196 | 0.208 |

**Table 2. Nested ANOVA of physiological characteristics between and within populations of *S. superba*.**

| Trait | Mean square | | | F-value | | Fst |
|---|---|---|---|---|---|---|
| | Inter-population | Intra-population | Error | Inter-population | Intra-population | |
| CHL | 4.145 | 1.249 | 0.011 | 367.947** | 110.845** | 0.766 |
| MDA | 185.916 | 379.78 | 30.230 | 6.150** | 12.563** | 0.275 |
| Pro | 1012522.754 | 234577.900 | 72711.850 | 13.925** | 3.226** | 0.754 |
| SP | 64.303 | 63.394 | 4.749 | 13.539** | 13.348** | 0.466 |
| SS | 0.004 | 0.001 | 0.0001 | 41.933** | 13.586** | 0.737 |

** indicates significance at $p < 0.01$.

## SSR-based genetic diversity analysis

Twenty polymorphic SSR loci were used to assess genetic diversity among eight populations of *S. superba*. Summary statistics for the main diversity parameters are shown in Table 3. Across all loci, Na ranged from 6 to 30, and Ne varied between 1.634 and 16.527. I values ranged from 0.796 to 3.011, Ho from 0.182 to 0.810, He from 0.554 to 0.901, F values ranged from 0.061 to 0.691, and Hs from 0.738 to 0.926. PIC values indicated high polymorphism levels at all loci, ranging from 0.363 to 0.936. Fis ranged from −0.041 to 0.481, and Fit from 0.060 to 0.531. Fst values, reflecting inter-population genetic differentiation, ranged from 0.047 to 0.168, while gene flow (Nm) values varied between 1.237 and 5.087.

Among the twenty loci, SS36 exhibited the highest levels of genetic diversity, with the largest Na (30), Ne (16.527), I (3.011), He (0.939), and PIC (0.936). It also showed the highest Ho (0.810) together with a low F (0.138), indicating abundant and evenly distributed alleles at this locus. At the opposite extreme, SS24 showed the lowest diversity (Na = 6, Ne = 1.634, I = 0.796), the lowest Ho (0.182), and a high F (0.531), reflecting a pronounced heterozygote deficit. The strongest deficit was at SS12 (F = 0.691; Fis = 0.481; Fit = 0.531), suggesting substantial inbreeding or a Wahlund effect at this locus. By contrast, SS32 and SS22 displayed negative Fis values (−0.041, −0.018) and high Ho (0.793), evidencing heterozygote excess. SS10 also showed high Ho (0.711) with the lowest positive Fis (0.017), suggesting near-random mating within populations. Regarding inter-population structure, the greatest differentiation occurred at SS21 (Fst = 0.168) with the lowest estimated gene flow (Nm = 1.237), whereas SS20 recorded the smallest Fst (0.047) and the highest Nm (5.087), implying extensive allele exchange among populations at this locus.

**Table 3. Statistical values of microsatellite markers in 122 samples of 8 S. superba populations.**

| Locus | Na | Ne | I | Ho | He | F | Hs | PIC | Fis | Fit | Fst | Nm |
|---|---|---|---|---|---|---|---|---|---|---|---|---|
| SS02 | 19 | 3.235 | 1.654 | 0.545 | 0.691 | 0.211 | 0.853 | 0.649 | 0.125 | 0.239 | 0.130 | 1.673 |
| SS05 | 18 | 4.589 | 2.075 | 0.504 | 0.782 | 0.355 | 0.857 | 0.767 | 0.336 | 0.437 | 0.152 | 1.396 |
| SS08 | 11 | 3.982 | 1.715 | 0.636 | 0.749 | 0.150 | 0.866 | 0.714 | 0.106 | 0.224 | 0.132 | 1.645 |
| SS10 | 10 | 4.784 | 1.771 | 0.711 | 0.791 | 0.101 | 0.854 | 0.762 | 0.017 | 0.105 | 0.090 | 2.539 |
| SS11 | 18 | 8.487 | 2.392 | 0.445 | 0.882 | 0.495 | 0.896 | 0.872 | 0.316 | 0.404 | 0.129 | 1.695 |
| SS12 | 15 | 8.087 | 2.304 | 0.271 | 0.876 | 0.691 | 0.887 | 0.865 | 0.481 | 0.531 | 0.097 | 2.318 |
| SS13 | 14 | 6.851 | 2.177 | 0.603 | 0.854 | 0.294 | 0.903 | 0.839 | 0.108 | 0.176 | 0.076 | 3.053 |
| SS16 | 11 | 5.078 | 1.888 | 0.316 | 0.803 | 0.606 | 0.854 | 0.779 | 0.359 | 0.465 | 0.165 | 1.264 |
| SS18 | 24 | 9.743 | 2.632 | 0.587 | 0.897 | 0.346 | 0.918 | 0.889 | 0.240 | 0.310 | 0.091 | 2.496 |
| SS19 | 15 | 8.571 | 2.270 | 0.620 | 0.883 | 0.298 | 0.896 | 0.872 | 0.296 | 0.361 | 0.092 | 2.466 |
| SS20 | 14 | 6.534 | 2.096 | 0.642 | 0.847 | 0.242 | 0.911 | 0.829 | 0.094 | 0.136 | 0.047 | 5.087 |
| SS21 | 13 | 2.241 | 1.298 | 0.355 | 0.554 | 0.358 | 0.805 | 0.528 | 0.122 | 0.270 | 0.168 | 1.237 |
| SS22 | 16 | 6.573 | 2.166 | 0.793 | 0.848 | 0.064 | 0.891 | 0.833 | −0.018 | 0.060 | 0.077 | 3.006 |
| SS23 | 12 | 7.086 | 2.173 | 0.483 | 0.859 | 0.438 | 0.870 | 0.844 | 0.357 | 0.460 | 0.160 | 1.311 |
| SS24 | 6 | 1.634 | 0.796 | 0.182 | 0.388 | 0.531 | 0.738 | 0.363 | 0.297 | 0.364 | 0.096 | 2.364 |
| SS27 | 13 | 5.858 | 2.004 | 0.645 | 0.829 | 0.223 | 0.851 | 0.811 | 0.210 | 0.315 | 0.133 | 1.628 |
| SS30 | 21 | 10.148 | 2.566 | 0.711 | 0.901 | 0.212 | 0.903 | 0.894 | 0.147 | 0.240 | 0.109 | 2.035 |
| SS32 | 20 | 6.443 | 2.279 | 0.793 | 0.845 | 0.061 | 0.898 | 0.831 | −0.041 | 0.082 | 0.118 | 1.873 |
| SS36 | 30 | 16.527 | 3.011 | 0.810 | 0.939 | 0.138 | 0.926 | 0.936 | 0.087 | 0.173 | 0.094 | 2.410 |
| SS42 | 15 | 7.322 | 2.229 | 0.636 | 0.863 | 0.263 | 0.905 | 0.850 | 0.320 | 0.370 | 0.074 | 3.149 |
| **Mean** | 15.75 | 6.689 | 2.075 | 0.564 | 0.804 | 0.304 | 0.874 | 0.786 | 0.198 | 0.286 | 0.111 | 2.232 |

Notably, several loci such as SS36, SS30, SS18, and SS19 exhibited both high expected heterozygosity (He ≥ 0.883) and high polymorphism information content (PIC ≥ 0.872), confirming that the marker panel was highly informative for detecting genetic variation in *S. superba*.

To visualize the genetic relationships among individuals, an SSR-based dendrogram was constructed from the multilocus genotypes of 122 individuals (Fig 1). The dendrogram revealed several major lineages and partial population-level aggregation. In particular, individuals from the HGY and HBN populations tended to cluster closely with several HCW and HTHL samples, whereas individuals from the large HBB and HQZ populations were distributed across multiple branches, indicating admixture and weak geographic structuring. Overall, this clustering pattern agrees with the moderate genetic differentiation (mean Fst = 0.111) and relatively high gene flow (mean Nm = 2.232) estimated from the SSR loci (Table 3).

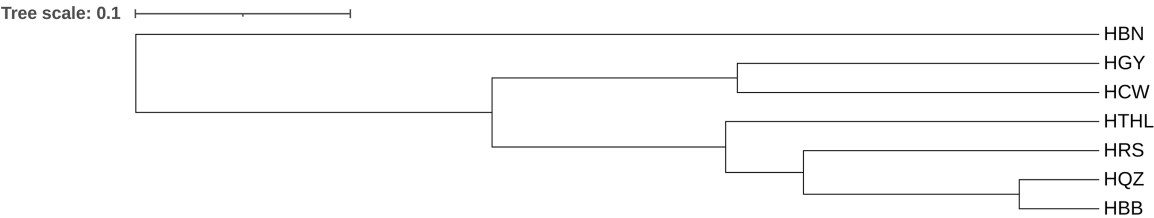

**Fig 1. UPGMA dendrogram of the eight populations constructed from SSR-based genetic distances.**

## Genetic differentiation among populations

A population-level analysis of genetic variation was conducted for eight natural populations of *S. superba* using 20 SSR loci (Table 4). The average number of individuals (N) per population ranged from 3.000 (HBN) to 45.600 (HBB). The mean Na per population ranged from 3.850 (HBN) to 11.100 (HBB), while the number of effective alleles (Ne) ranged from 3.268 (HBN) to 6.506 (HTHL). Shannon's information index (I) varied between 1.200 and 1.957, with the highest value observed in HTHL and the lowest in HBN.

Ho values ranged from 0.522 (HBB) to 0.717 (HBN), and He ranged from 0.644 to 0.818. The uHe ranged from 0.736 (HQZ) to 0.856 (HTHL). The inbreeding coefficient (F) varied from −0.113 (HBN) to 0.333 (HTHL), suggesting varying levels of deviation from Hardy–Weinberg equilibrium (HWE). The proportion of loci deviating from HWE across populations ranged from 5.000% (HGY) to 80.000% (HQZ), with most populations showing moderate to high levels of deviation.

The genetic relationships among the eight populations were further evaluated based on Nei's genetic distance and genetic identity (Table 5). The genetic distance values ranged from 0.074 to 1.006, and genetic identity coefficients varied from 0.366 to 0.929. Generally, a greater genetic distance corresponded to lower genetic identity, indicating a more distant

**Table 4. Summary of genetic statistics for S. superba at population level.**

| Pop | | N | Na | Ne | I | Ho | He | uHe | F | Percentage of Deviation from HWE Site (%) |
|---|---|---|---|---|---|---|---|---|---|---|
| HBB | Mean | 45.600 | 11.100 | 5.382 | 1.811 | 0.522 | 0.750 | 0.758 | 0.313 | 75.000 |
| | SE | 0.311 | 0.846 | 0.616 | 0.116 | 0.049 | 0.037 | 0.037 | 0.053 | |
| HBN | Mean | 3.000 | 3.850 | 3.268 | 1.200 | 0.717 | 0.644 | 0.773 | −0.113 | ns |
| | SE | 0.000 | 0.244 | 0.271 | 0.082 | 0.061 | 0.035 | 0.042 | 0.069 | |
| HCW | Mean | 7.950 | 6.900 | 5.134 | 1.722 | 0.590 | 0.784 | 0.837 | 0.255 | 15.000 |
| | SE | 0.050 | 0.458 | 0.391 | 0.071 | 0.043 | 0.015 | 0.016 | 0.047 | |
| HGY | Mean | 5.000 | 4.950 | 3.794 | 1.434 | 0.630 | 0.722 | 0.802 | 0.137 | 5.000 |
| | SE | 0.000 | 0.185 | 0.187 | 0.047 | 0.057 | 0.016 | 0.018 | 0.070 | |
| HQZ | Mean | 32.500 | 8.800 | 4.738 | 1.662 | 0.563 | 0.724 | 0.736 | 0.231 | 80.000 |
| | SE | 0.328 | 0.631 | 0.448 | 0.115 | 0.051 | 0.043 | 0.044 | 0.054 | |
| HRS | Mean | 14.750 | 7.900 | 5.649 | 1.809 | 0.641 | 0.795 | 0.823 | 0.205 | 60.000 |
| | SE | 0.123 | 0.481 | 0.423 | 0.080 | 0.044 | 0.022 | 0.023 | 0.043 | |
| HTHL | Mean | 11.000 | 9.150 | 6.506 | 1.957 | 0.550 | 0.818 | 0.856 | 0.333 | 55.000 |
| | SE | 0.000 | 0.617 | 0.623 | 0.084 | 0.049 | 0.018 | 0.019 | 0.055 | |

**Table 5. Analysis of genetic distance and genetic consistency.**

| Pop | HBB | HBN | HCW | HGY | HQZ | HRS | HTHL |
|---|---|---|---|---|---|---|---|
| HBB | *** | 0.366 | 0.577 | 0.613 | 0.929 | 0.767 | 0.731 |
| HBN | 1.006 | *** | 0.442 | 0.371 | 0.368 | 0.438 | 0.479 |
| HCW | 0.549 | 0.816 | *** | 0.714 | 0.513 | 0.592 | 0.605 |
| HGY | 0.490 | 0.991 | 0.336 | *** | 0.558 | 0.568 | 0.531 |
| HQZ | 0.074 | 0.999 | 0.667 | 0.583 | *** | 0.753 | 0.700 |
| HRS | 0.265 | 0.825 | 0.524 | 0.566 | 0.284 | *** | 0.690 |
| HTHL | 0.313 | 0.736 | 0.503 | 0.633 | 0.357 | 0.371 | *** |

The upper triangular matrix is genetic consistency, and the lower triangular matrix is genetic distance. *** Indicates that the genetic distance is 0 and the genetic consistency is 1.

genetic relationship between the populations. Conversely, populations with smaller genetic distances exhibited higher genetic identities, reflecting closer genetic affinities.

As shown in Table 5, the smallest genetic distance (0.074) was observed between HBB and HQZ, with a corresponding genetic identity of 0.929. This indicated that the genetic relationship between HBB and HQZ was the closest among all population pairs. In contrast, the greatest genetic distance was found between HBN and HBB (1.006), and the genetic identity between these two populations was the lowest at 0.366, suggesting the greatest genetic divergence between them. Other closely related population pairs included HCW and HGY, while more divergent relationships were seen between HBN and several other populations, such as HQZ and HRS.

## Genetic relationship and population structure analysis

PCoA was used to visualize the spatial genetic distribution among individuals and to evaluate the degree of genetic divergence between populations. The PCoA plot based on 20 SSR markers from 122 individuals of *S. superba* is shown in Fig 2. The first principal coordinate (PCoA1) and the second principal coordinate (PCoA2) explained 10.271% and 4.764% of the total genetic variation, respectively. Individuals from different populations showed partial overlap in the coordinate space, indicating genetic admixture among populations. However, individuals from populations HBN and HTHL were relatively distant from the rest, suggesting a certain degree of genetic differentiation.

The optimal number of genetic clusters (K) was inferred using the ΔK method based on Bayesian clustering (Fig 3). The peak of ΔK was observed at K = 5, indicating that the most likely number of genetic groups was five. The STRUCTURE bar plot at K = 5 (Fig 4) showed that most individuals displayed mixed ancestry components, with varying degrees of membership across the five clusters. Some populations, such as HBB and HQZ, showed a relatively high proportion of shared ancestry, while others, including HBN and HTHL, exhibited distinct genetic compositions.

## Association between phenotypic traits and SSR markers

A mixed linear model (MLM) was employed to detect associations between SSR loci and physiological traits across all individuals, incorporating the population structure matrix (Q matrix at K = 5) as a covariate. The results of the MLM-based

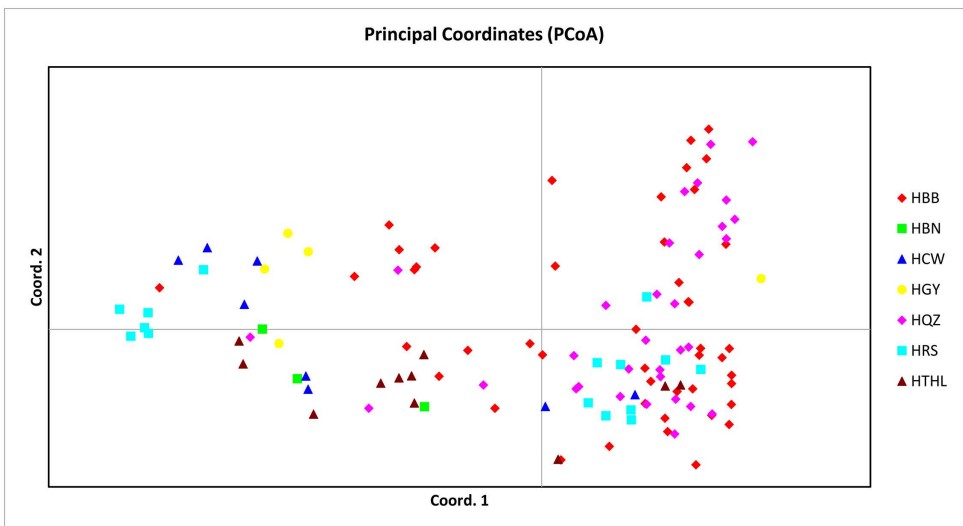

**Fig 2. Principal coordinate analysis (PCoA) of 122 *S. superba* based on 20 SSR markers.** The different colors and shapes represent different study populations. The first and second axes explained10.271% and 4.764% of the genetic similarities among populations, respectively.

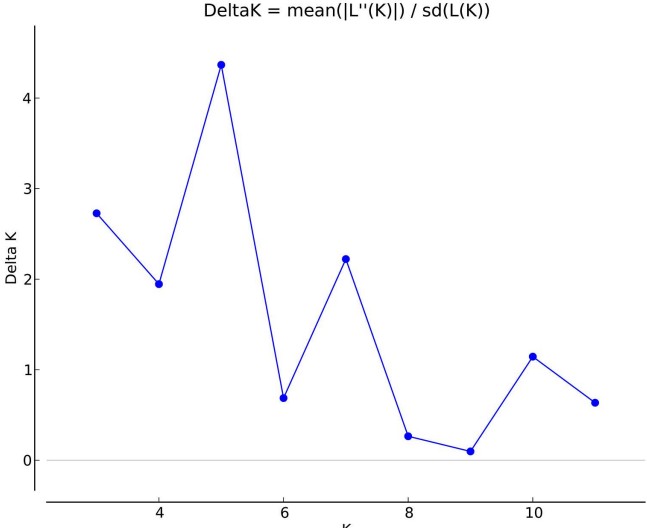

**Fig 3. Delta K values for estimating the optimal number of clusters.**

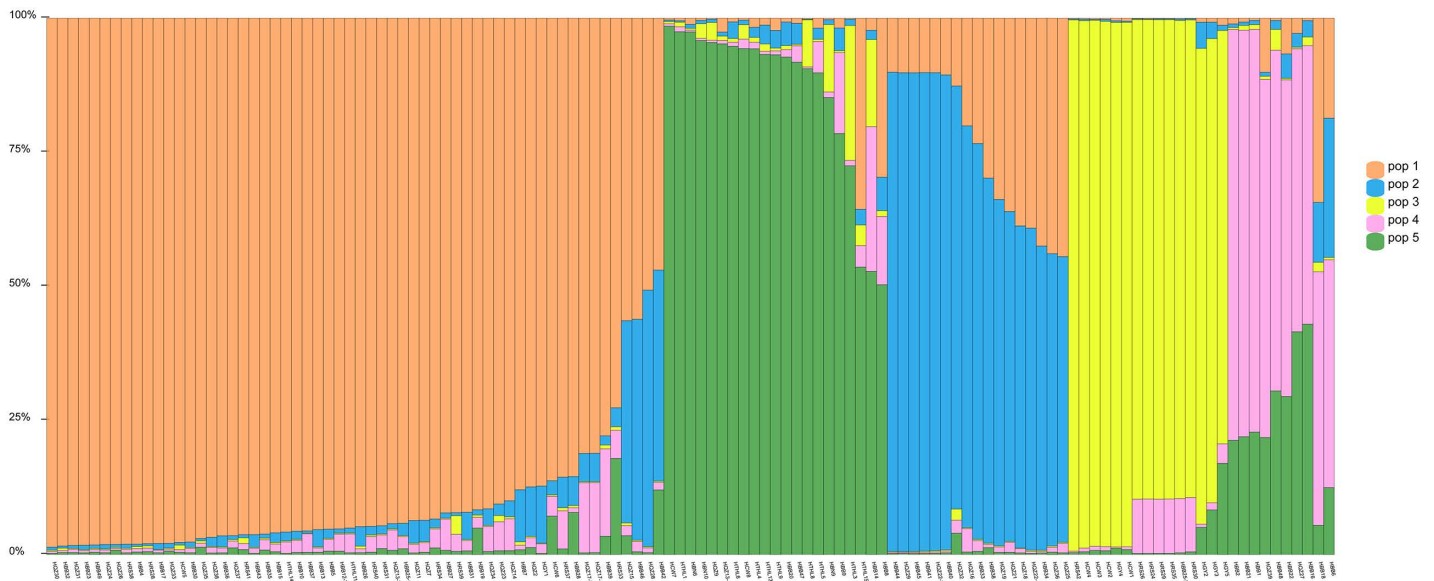

**Fig 4. K = 5 cluster STRUCTURE ancestral proportion bar chart.** Each individual is represented as a line segment, divided vertically by different colours, representing the proportion of ancestors estimated by the individual in each cluster. The number below the figure represents the sampling population.

association analysis are summarized in Table 6. A total of 14 significant marker-trait associations ($p < 0.05$) were identified, with $R^2$ values ranging from 0.2129 to 0.6938, indicating moderate to strong contributions of individual loci to trait variation.

For CHL, six SSR loci (SS08, SS10, SS16, SS24, SS27, and SS30) were significantly associated, among which SS30 showed the highest explanatory power ($R^2 = 0.6938$, $p = 0.0166$). The SSR marker SS27 also showed strong association

**Table 6. Trait-marker association analysis based on MLM model.**

| Trait | SSR | F value | P | $R^2$ |
|---|---|---|---|---|
| CHL | SS08 | 1.7860 | 0.0344 | 0.4022 |
| | SS10 | 2.4079 | 0.0036 | 0.4286 |
| | SS16 | 1.7854 | 0.0345 | 0.4021 |
| | SS24 | 2.2654 | 0.0304 | 0.2129 |
| | SS27 | 2.2852 | 0.0038 | 0.4990 |
| | SS30 | 1.9095 | 0.0166 | 0.6938 |
| MDA | SS05 | 1.6886 | 0.0436 | 0.4656 |
| | SS19 | 1.8529 | 0.0192 | 0.6385 |
| | SS24 | 4.1017 | 0.0004 | 0.3295 |
| | SS30 | 1.6688 | 0.0453 | 0.6732 |
| SP | SS02 | 1.7228 | 0.0495 | 0.3645 |
| | SS10 | 2.5955 | 0.0019 | 0.4398 |
| SS | SS05 | 2.0338 | 0.0102 | 0.4801 |
| | SS08 | 1.8123 | 0.0310 | 0.3864 |
| | SS32 | 2.1049 | 0.0062 | 0.6266 |
| | SS42 | 2.5255 | 0.0010 | 0.5706 |

with CHL ($R^2$=0.4990). For MDA, four loci (SS05, SS19, SS24, and SS30) were detected as significantly associated. The strongest association was observed at SS30 ($R^2$=0.6732, $p$=0.0453), followed by SS19 ($R^2$=0.6385, $p$=0.0192). For SP, significant associations were observed at SS02 and SS10, with SS10 showing higher explanatory power ($R^2$=0.4398, $p$=0.0019). For SS, four loci (SS05, SS08, SS32, and SS42) were significantly associated. Among them, SS32 and SS42 exhibited high explanatory power with $R^2$ values of 0.6266 and 0.5706, respectively.

## Discussion

### Physiological and genetic diversity of *S. superba*

This study revealed substantial physiological and genetic variability across eight natural populations of *S. superba*. Five physiological traits (CHL, MDA, Pro, SP, SS) displayed significant among-population differences, with HTHL and HBB exhibiting the highest coefficients of variation (0.31, 0.30). Particularly, MDA showed pronounced intra-population variability (CV > 0.8), indicating differential oxidative stress responses among individuals under local environmental conditions.

On the molecular side, 20 SSR loci yielded high polymorphism, with a mean Na of 15.75, mean Ne of 6.689, mean He of 0.804, and mean PIC of 0.786, all of which reflect a rich allelic reservoir. These values are comparable to those reported in *S. superba* plus-tree germplasm analyzed using SSRs, where the average Na reached 19 and He ranged up to 0.945 [14]. In fact, the values here are very similar to those found in *Quercus liaotungensis* (He=0.80) [33], which puts *S. superba* toward the high end of genetic diversity for broad-leaved trees. Compared to other woody species such as *Paeonia decomposita* (He=0.492) [34] or *Populus tomentosa* (PIC=0.56) [35], *S. superba* exhibited notably higher genetic diversity.

Within-marker variation was considerable: loci SS36 and SS30 had PIC of 0.94 and 0.89, respectively; loci SS10 and SS22 had Ho exceeding 0.70. In contrast, SS11 and SS24 showed large Ho-He discrepancies, hinting at possible null alleles or inbreeding effects. At the population level, HTHL and HBB maintained the highest diversity (Na > 9.0, He > 0.75), aligning with their large physiological trait variability. Conversely, HBN and HGY exhibited lower allelic richness, possibly resulting from geographic isolation or reduced effective population sizes. Several populations (HQZ) had up to 80% loci deviating from Hardy-Weinberg equilibrium, suggesting non-random mating or substructuring. Similar heterozygote deficits have also been observed in Castanopsis populations affected by inbreeding or substructure [36].

Importantly, a correlation was observed between high physiological variability (HTHL and HBB) and elevated genetic diversity, supporting the hypothesis that genetic variation enhances plant adaptive capacity. A similar pattern has been reported in *Pinus koraiensis*, where populations with greater phenotypic variability also exhibited higher genetic diversity indices (He = 0.528, Shannon's I = 1.103) [37]. Overall, the combined analysis of physiological traits and SSR-based genotyping provides strong evidence for extensive adaptive and genetic diversity in *S. superba*. Such diversity is crucial for the species' resilience to environmental changes and offers valuable theoretical support for breeding and germplasm conservation efforts.

## Population differentiation and genetic structure

The SSR-based dendrogram (Fig 1) showed clear population clustering patterns. HBN formed a distinct branch separated from the other provenances, suggesting a relatively divergent genetic background. The remaining populations grouped into two major lineages, with HGY and HCW clustering closely, and HQZ and HBB forming the tightest subcluster, indicating high genetic similarity and possible gene exchange or shared ancestry. Overall, the dendrogram supports the presence of both broadly shared genetic backgrounds among populations and localized differentiation in specific provenances, which is relevant for germplasm conservation and targeted utilization.

The genetic structure of *S. superba* populations was assessed using multiple complementary approaches, including fixation indices, analysis of molecular variance (AMOVA), PCoA, and structure-based clustering. Overall, the observed genetic differentiation among populations was moderate, with Fst values ranging from 0.08 to 0.12 across loci, and estimated gene flow (Nm) values consistently exceeding 1. These results suggest a balance between genetic divergence and inter-population connectivity [38]. The AMOVA results further supported this observation. Of the total genetic variance, 68.53% was attributed to within-individual variation, 27.29% to differences among individuals within populations, and only 4.18% to differences among populations (Table 7). This pattern is consistent with typical outcrossing tree species and reflects the high level of gene flow across populations. Similar findings were reported in *Camellia nitidissima* var. *phaeopubisperma* [39], which showed Fst ≈ 0.073 and Nm ≈ 7.37, indicating frequent inter-population gene exchange despite moderate structuring. The PCoA based on SSR markers revealed partial separation of individuals from different populations along the first two axes, which explained 10.27% and 4.76% of the total genetic variance, respectively. In support of this, similar patterns have been observed in other tree species, such as *Myrica esculenta* [40] in the Western Himalayas and *Campomanesia xanthocarpa* [41] in Brazil, both showing high genetic diversity and structure shaped by connectivity and landscape factors. Notably, individuals from populations HBN and HTHL were more distant from the main cluster, indicating a certain degree of genetic uniqueness. These spatial patterns were generally in agreement with the structure analysis, which identified five distinct genetic clusters (K = 5) based on the ΔK method. While most individuals exhibited admixture from multiple clusters, some populations such as HBN and HTHL showed relatively pure ancestry, reflecting their genetic differentiation. A comparable pattern was documented in *S. superba* by Bai [17], who identified seven genetic clusters using SSR and SNP data, with STRUCTURE and PCoA revealing overlapping but distinct groupings among populations. The presence of clear substructure despite moderate differentiation underscores the importance of considering both geographic isolation and local selective pressures in shaping genetic patterns [42].

**Table 7. AMOVA of genetic variation within and among groups of *S. superba*.**

| Source of Variation | df | SS | Est. Var. | Variation (%) |
| --- | --- | --- | --- | --- |
| Among Pops | 6 | 122.65 | 0.34 | 4.18 |
| Among Indiv | 114 | 1149.17 | 2.23 | 27.29 |
| Within Indiv | 121 | 679.00 | 5.61 | 68.53 |
| Total | 241 | 1950.82 | 8.19 | 100.00 |

Collectively, the clustering pattern observed in genetic distance, PCoA, and STRUCTURE showed an evident geographic signal, suggesting that spatial proximity and regional connectivity contributed to the genetic structure of *S. superba* in southeastern Guangxi. Notably, the closest pair (HBB–HQZ) is also geographically adjacent (S1 Table), which is consistent with an isolation-by-distance pattern where genetic similarity decays with increasing geographic distance [43,44]. In addition, the eastern provenances (HBN, HTHL, and HCW) are clustered within a narrow coordinate range (111.48–111.69°E; 23.86–23.95°N), and their genetic similarity may be largely driven by gene flow among nearby populations within this region. Meanwhile, several populations still displayed admixture rather than strict geographic partitioning, which may be biologically explained by the predominantly outcrossing mating system and pollen-mediated gene flow reported for *S. superba*, potentially weakening strict geographic clustering at the population level [45]. Similar landscape-induced genetic structure has been documented in other species such as *Juglans sigillata* [46], where long-term habitat fragmentation and topographic barriers shaped differentiation despite ongoing gene flow.

In summary, the genetic structure of *S. superba* is shaped by a combination of gene flow, local adaptation, and geographic context. Moderate Fst values and high Nm indicate substantial genetic connectivity, yet structure and PCoA reveal distinct substructures among specific populations. These findings suggest that when using different *S. superba* populations for breeding and afforestation, their degree of genetic differentiation should be carefully considered to meet breeding goals and achieve the highest economic value.

## Association between SSR Markers and Physiological Traits

In this study, mixed linear model (MLM) analysis integrating both population structure (Q) and kinship (K) matrices identified significant associations between several SSR loci and physiological traits, notably CHL, MDA, Pro, SP, and SS. Among these markers, SS30, SS32, and SS27 were consistently associated with multiple traits and explained substantial portions of phenotypic variance, with $R^2$ values ranging from approximately 0.12 to 0.20 (Table 6).

This pattern of pleiotropic SSR-trait associations has also been observed in other tree species. For example, in *Paeonia rockii* [47], 41 EST-SSR markers were linked to multiple yield-related traits and exhibited pleiotropic associations, facilitating discovery of genomic regions for marker-assisted breeding. Similarly, in Festuca arundinacea [48], SSR markers were significantly associated with physiological traits such as chlorophyll content under heat stress, with $R^2$ values exceeding 0.10, demonstrating utility in breeding for abiotic stress tolerance.

Importantly, the markers we identified (SS30 and SS32) not only exhibit statistical association with phenotypes, but also possess high polymorphism (PIC > 0.89) and allele richness, which are critical characteristics for reliable marker development and deployment. This combination of high informativeness and functional relevance enhances their value for early selection in *S. superba* breeding programs.

Association mapping in forest tree species commonly identifies a modest number of SSRs significantly linked to complex traits, often explaining ≤20% of variance. For instance, marker–trait studies in maize and rice reported average $R^2$ values between 10–15% using Q + K MLM frameworks [49]. Our findings are consistent with this trend and highlight the polygenic nature of physiological traits in perennial tree species.

From a practical perspective, SSR markers such as SS30 and SS32 represent promising candidates for marker assisted selection (MAS) in *S. superba*. They could be used to screen seedlings for desirable physiological profiles, such as increased CHL and reduced MDA, thereby enabling early-stage selection and accelerating breeding cycles. Additionally, the genomic regions linked with these SSR loci warrant further investigation via fine mapping or transcriptome analysis to verify candidate genes. In summary, the significant SSR associations identified here extend the functional genetic basis of physiological trait variation in *S. superba*. These markers hold potential for MAS in future breeding and conservation programs aimed at enhancing adaptive and functional traits.

## Implications for conservation and future research

The results of this study provide valuable insights into the conservation and utilization of *S. superba* genetic resources. The high genetic diversity observed within populations, which is reflected by elevated values of He and allele richness, suggests that this species maintains a robust genetic foundation to cope with environmental change. Populations such as HTHL and HBB, which exhibit both high molecular and physiological variability, should be prioritized in conservation efforts as they represent reservoirs of adaptive potential. Previous research has emphasized the importance of conserving genetically diverse populations to maximize evolutionary resilience, as demonstrated in species such as *Camellia nitidissima* [50]and Quercus variabilis [51].

Despite moderate levels of population differentiation indicated by Fst and STRUCTURE analyses, some populations such as HBN and HQZ showed distinct genetic profiles, possibly reflecting geographic isolation or site-specific adaptation. These distinct populations deserve consideration as conservation units to safeguard regionally important alleles. Similar conservation strategies have been suggested for *Paeonia decomposita* [34]and other endemic woody plants, in which geographically structured genetic lineages are preserved through both in situ and ex situ approaches [52]. For *S. superba*, adopting such strategies would be more effective when genetic structure data are combined with ecological information to delineate evolutionary significant units more accurately.

The identification of functionally associated SSR loci further enhances the practical application of this study. Markers such as SS30 and SS32, which are significantly correlated with chlorophyll content, malondialdehyde concentration, and other stress-related traits, offer promising tools for molecular-assisted breeding. Their high polymorphism and phenotypic explanatory power suggest they could be implemented in early-generation selection to accelerate the development of stress-resilient genotypes [53]. This approach is increasingly employed in tree breeding programs, as demonstrated in *Paeonia rockii*, where SSR markers were associated with multiple yield-related traits [47], and in Populus deltoides, where SSR markers have been linked to water-use and nitrogen-use efficiency traits to guide elite line selection [54].

Future research should aim to validate these associations through fine mapping, transcriptomic analysis, and linkage mapping [55]. Incorporating functional markers into broader genome-wide association studies (GWAS) or environmental association models would further elucidate the genetic basis of adaptive traits [56]. Over time, we need to watch how the mix of genes in populations shifts with climate change, habitat loss, and other human activities [57]. Integrating SSRs with functional markers (EST-SSRs) enables more precise identification of associations between genetic loci and adaptive traits in plants, thereby strengthening molecular breeding and germplasm conservation efforts [58].

## Conclusions

The provenances of *S. superba* evaluated in this study exhibited substantial genetic variation and rich genetic diversity, indicating a relatively broad genetic basis in southeastern Guangxi. Among them, HTHL and HBB were identified as the most promising populations for further exploitation and utilization because they showed high genetic diversity and strong adaptive potential, providing valuable germplasm for future breeding, conservation, and ecological restoration programs. In addition, to avoid potential loss of unique characteristics caused by genetic admixture among populations, provenances with relatively distinct genetic backgrounds (e.g., HBN and HTHL) should be considered as priority units for targeted conservation, such as establishing resource nurseries. Furthermore, SSR loci SS30 and SS32 exhibited significant associations with key physiological traits, suggesting their potential value as candidate functional markers for early screening and marker-assisted selection in *S. superba*.

## Supporting information

**S1 Table. The longitude and latitude of the 8 S. superba provenance collection sites.**
(DOCX)

**S2 Table. Characteristics of 20 pairs of polymorphic SSR primers in *S. superba*.**
(XLSX)

**S1 File. Raw data.**
(ZIP)

## Author contributions

**Data curation:** Jiazhe Liu, Yu Zhong.

**Formal analysis:** Jiazhe Liu, Yu Zhong.

**Funding acquisition:** Zhangqiang Tan.

**Project administration:** Wenhui Shen, Zhangqiang Tan.

**Resources:** Wenhui Shen.

**Supervision:** Zhangqiang Tan.

**Visualization:** Yu Zhong.

**Writing – original draft:** Jiazhe Liu.

**Writing – review & editing:** Jiazhe Liu.

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
