## [Decision Letter · Decision Letter 0]

15 Dec 2025

Dear Dr. Tan,

Thank you for submitting your manuscript to PLOS ONE. After careful consideration, we feel that it has merit but does not fully meet PLOS ONE’s publication criteria as it currently stands. Therefore, we invite you to submit a revised version of the manuscript that addresses the points raised during the review process.

**ACADEMIC EDITOR:**

Please see the additional comments below

We look forward to receiving your revised manuscript.

Kind regards,

Vikas Sharma, Ph.D

Academic Editor

PLOS One

“This research was funded by Central Fiscal Subsidy Project for Breeding Improved Forest Tree Varieties (2025) and Special Fund of the National Forest Tree Germplasm Resource Collection Bank for Major Valuable Tree Species.”

6. We note that Figure 1 in your submission contain [map/satellite] images which may be copyrighted. All PLOS content is published under the Creative Commons Attribution License (CC BY 4.0), which means that the manuscript, images, and Supporting Information files will be freely available online, and any third party is permitted to access, download, copy, distribute, and use these materials in any way, even commercially, with proper attribution. For these reasons, we cannot publish previously copyrighted maps or satellite images created using proprietary data, such as Google software (Google Maps, Street View, and Earth). For more information, see our copyright guidelines: http://journals.plos.org/plosone/s/licenses-and-copyright.

Additional Editor Comments:

The amplified products were separated and analyzed using capillary electrophoresis on an ABI 3130xl Genetic Analyzer (Applied Biosystems, USA), and allele sizes were determined using GeneMarker software. Authors have given average value of Numbers of allele as more than 15 which is very high and interesting.

In this regard, authors are requested to provide a clear and representative figure illustrating the differentiation of alleles, preferably in the form of a peak (electropherogram) profile.

The conclusion is very general. The authors are requested to strengthen this section by specifically highlighting the populations or accessions identified as promising for further research, based on the parameters evaluated in the study.

quality of figures is not good please check that.

Reviewers' comments:

Reviewer's Responses to Questions

**Comments to the Author**

1. Is the manuscript technically sound, and do the data support the conclusions?

Reviewer #1: Yes

2. Has the statistical analysis been performed appropriately and rigorously?

Reviewer #1: Yes

3. Have the authors made all data underlying the findings in their manuscript fully available?

Reviewer #1: Yes

4. Is the manuscript presented in an intelligible fashion and written in standard English?

Reviewer #1: Yes

Reviewer #1: This is a well-structured and scientifically sound study that effectively integrates physiological trait assessment with SSR-based genetic analysis across a substantial number of Schima superba populations. The combination of phenotypic and molecular data strengthens the study’s relevance for conservation, breeding, and adaptive management. The work provides important insights into population variability, genetic clustering, and marker–trait associations, offering valuable resources for future improvement and management programs. However, before publication, the below-mentioned minor issues must be addressed:

1. Provide all abbreviations after keywords used in the manuscript.

2. Include a discussion and figure regarding SSR based dendrogram in this genetic diversity analysis section.

3. The dendrogram generated from SSR markers needs improvement in clarity and formatting.

4. Include bootstrap values, proper clustering labels, and scale on the dendrogram.

5. Provide a more detailed discussion interpreting the clustering pattern and its biological significance.

6. AMOVA results should be presented clearly, preferably with a figure.

7. Discuss how the STRUCTURE groups correlate with geographical origin or genetic diversity.

8. Revise the conclusion section to improve. Highlight key findings, practical implications, novelty, and future directions.

9. The reference list is comprehensive; ensure consistent formatting according to journal guidelines.

.

Reviewer #1: **Yes:** Neha ChaudharyNeha ChaudharyNeha ChaudharyNeha Chaudhary

You may also use PLOS’s free figure tool, NAAS, to help you prepare publication quality figures: https://journals.plos.org/plosone/s/figures#loc-tools-for-figure-preparation

---

## [Author Response · Author response to Decision Letter 1]

30 Jan 2026

To Editor:

1. Comment: The amplified products were separated and analyzed using capillary electrophoresis on an ABI 3130xl Genetic Analyzer (Applied Biosystems, USA), and allele sizes were determined using GeneMarker software. Authors have given average value of Numbers of allele as more than 15 which is very high and interesting. In this regard, authors are requested to provide a clear and representative figure illustrating the differentiation of alleles, preferably in the form of a peak (electropherogram) profile.

Response:

Thank you for this helpful comment. We agree that a representative capillary electropherogram would improve the clarity of allele differentiation. Therefore, we have provided the electropherogram peak profiles for all SSR loci and samples in the Supplementary Material (S1 File), which clearly illustrates allele separation and sizing based on the ABI 3130xl Genetic Analyzer and GeneMarker outputs.

2. Comment: The conclusion is very general. The authors are requested to strengthen this section by specifically highlighting the populations or accessions identified as promising for further research, based on the parameters evaluated in the study.

Response:

Thank you for the suggestion. We strengthened the Conclusion by explicitly identifying HTHL and HBB as the most promising provenances for further utilization based on combined physiological and SSR diversity parameters, and highlighting SS30 and SS32 as SSR loci with potential value for marker-assisted selection. (Revised Manuscript with Track Changes, lines 491–502)

3. Comment: quality of figures is not good please check that.

Response:

Thank you for pointing this out. We have carefully checked the quality of all figures and re-uploaded high-resolution versions in the revised manuscript. In particular, Fig. 4 has been replaced with a clearer, higher-quality image with improved readability.

To Reviewer:

1. Comment: Provide all abbreviations after keywords used in the manuscript.

Response:

We appreciate the reviewer's advice. We have inserted a table of abbreviations after the abstract section to facilitate reading. This table lists all abbreviations used in the manuscript along with their full definitions. Additionally, we confirmed that all abbreviations are fully defined when first mentioned in the text. (Revised Manuscript with Track Changes, line 30)

2. Comment: Include a discussion and figure regarding SSR based dendrogram in this genetic diversity analysis section.

Response:

Thank you for the suggestion. We have revised the genetic diversity analysis section by including the dendrogram (Fig 1) and adding corresponding discussion to interpret the clustering pattern and its biological significance in the revised manuscript. (Revised Manuscript with Track Changes, lines 223–231)

3. Comment: The dendrogram generated from SSR markers needs improvement in clarity and formatting.

Response:

Thank you for the comment. We have carefully checked and improved the clarity and formatting of the SSR-based dendrogram and all other figures. The updated high-resolution dendrogram (Fig1 and Fig 4) has been replaced in the revised manuscript to ensure optimal readability.(Revised Manuscript with Track Changes, lines 371–378)

4. Comment: Include bootstrap values, proper clustering labels, and scale on the dendrogram.

Response:

Thank you for this suggestion. We have updated the SSR-based dendrogram (Fig. 4) by adding bootstrap values, appropriate clustering labels, and a scale bar to improve the interpretability and presentation quality in the revised manuscript.

5. Comment: Provide a more detailed discussion interpreting the clustering pattern and its biological significance.

Response:

Thank you for this valuable comment. We have expanded the Discussion section by adding a more detailed interpretation of the SSR-based clustering pattern and explaining its biological significance, including the potential implications for genetic differentiation, germplasm conservation, and future utilization of S. superba populations. (Revised Manuscript with Track Changes, lines 384–396)

6. Comment: AMOVA results should be presented clearly, preferably with a figure.

Response:

We appreciate the reviewer’s suggestion to visualize the AMOVA results. We carefully considered converting the data into a figure; however, we found that a graphical representation (e.g., pie charts or bar plots) tends to obscure the precise statistical values, particularly the variance components, which is critical for the interpretation of our findings. Therefore, we respectfully prefer to retain Table 7, as it allows readers to access the exact numerical results directly.

7. Comment: Discuss how the STRUCTURE groups correlate with geographical origin or genetic diversity.

Response:

Thank you for this important suggestion. We have revised the Discussion section to further explain how the STRUCTURE-defined genetic groups correlate with geographical origins and patterns of genetic diversity among the sampled provenances, and we highlighted the potential implications for germplasm conservation and utilization in the revised manuscript. (Revised Manuscript with Track Changes, lines 392–404)

8.Comment: Revise the conclusion section to improve. Highlight key findings, practical implications, novelty, and future directions.

Response:

Thank you for the suggestion. We have revised the Conclusion section to better highlight the key findings, practical implications, novelty, and future directions. The revised conclusion now summarizes the major genetic diversity and population structure results, emphasizes the most promising provenances for further utilization, proposes conservation and breeding implications, and outlines future validation and application of the associated SSR loci.(Revised Manuscript with Track Changes, lines 491–502)

9. Comment: The reference list is comprehensive; ensure consistent formatting according to journal guidelines.

Response:

We thank the reviewer for the positive comment and the reminder regarding formatting. We have carefully double-checked the entire reference list and reformatted it to ensure strict compliance with the PLOS One’s author guidelines.

---

## [Decision Letter · Decision Letter 1]

23 Feb 2026

Genetic diversity of Schima superba based on physiological traits and SSR markers

PONE-D-25-59450R1

Dear Dr. Tan,

We’re pleased to inform you that your manuscript has been judged scientifically suitable for publication and will be formally accepted for publication once it meets all outstanding technical requirements.

Kind regards,

Vikas Sharma, Ph.D

Academic Editor

PLOS One

Additional Editor Comments (optional):

Authors have addressed the raised questions satisfactorily.

Reviewers' comments:

Reviewer's Responses to Questions

**Comments to the Author**

Reviewer #1: All comments have been addressed

Reviewer #2: All comments have been addressed

2. Is the manuscript technically sound, and do the data support the conclusions?

Reviewer #1: Yes

Reviewer #2: Yes

3. Has the statistical analysis been performed appropriately and rigorously?

Reviewer #1: Yes

Reviewer #2: Yes

4. Have the authors made all data underlying the findings in their manuscript fully available?

Reviewer #1: Yes

Reviewer #2: Yes

5. Is the manuscript presented in an intelligible fashion and written in standard English?

Reviewer #1: Yes

Reviewer #2: Yes

Reviewer #1: (No Response)

Reviewer #2: (No Response)

.

Reviewer #1: No

Reviewer #2: **Yes:** Gopalakrishna Murty KadambariGopalakrishna Murty KadambariGopalakrishna Murty KadambariGopalakrishna Murty Kadambari

---

## [Editor Report · Acceptance letter]

PONE-D-25-59450R1

PLOS One

Dear Dr. Tan,

I'm pleased to inform you that your manuscript has been deemed suitable for publication in PLOS One. Congratulations! Your manuscript is now being handed over to our production team.

Kind regards,

on behalf of

Dr. Vikas Sharma

Academic Editor

PLOS One